# Health to Eat: A Smart Plate with Food Recognition, Classification, and Weight Measurement for Type-2 Diabetic Mellitus Patients’ Nutrition Control

**DOI:** 10.3390/s23031656

**Published:** 2023-02-02

**Authors:** Salaki Reynaldo Joshua, Seungheon Shin, Je-Hoon Lee, Seong Kun Kim

**Affiliations:** 1Department of Electronics, Information and Communication Engineering, Kangwon National University, Samcheok-si 25913, Republic of Korea; 2Department of Computer Engineering, Kangwon National University, Samcheok-si 25913, Republic of Korea; 3Department of Liberal Studies, Kangwon National University, Samcheok-si 25913, Republic of Korea

**Keywords:** smart plate, diabetes, artificial intelligence, image recognition, weight, nutrition

## Abstract

The management of type 2 diabetes mellitus (T2DM) is generally not only focused on pharmacological therapy. Medical nutrition therapy is often forgotten by patients for several reasons, such as difficulty determining the right nutritional pattern for themselves, regulating their daily nutritional patterns, or even not heeding nutritional diet recommendations given by doctors. Management of nutritional therapy is one of the important efforts that can be made by diabetic patients to prevent an increase in the complexity of the disease. Setting a diet with proper nutrition will help patients manage a healthy diet. The development of Smart Plate Health to Eat is a technological innovation that helps patients and users know the type of food, weight, and nutrients contained in certain foods. This study involved 50 types of food with a total of 30,800 foods using the YOLOv5s algorithm, where the identification, measurement of weight, and nutrition of food were investigated using a Chenbo load cell weight sensor (1 kg), an HX711 weight weighing A/D module pressure sensor, and an IMX219-160 camera module (waveshare). The results of this study showed good identification accuracy in the analysis of four types of food: rice (58%), braised quail eggs in soy sauce (60%), spicy beef soup (62%), and dried radish (31%), with accuracy for weight and nutrition (100%).

## 1. Introduction

Nutrition is an important component that contributes to the continuation of the growth process. Children require nutrients such as protein, carbohydrates, fats, minerals, vitamins, and water during their growth and development period. If these requirements are not met, the process of further growth and development may be hampered [1]. Nutrients function as a basic material for the formation and repair of body cell tissue; as a protector and regulator of body temperature; and as a source of energy for organ function, motion, and physical function. Nutrients are elements required by the body for its processes and functions [2]. Energy is obtained from a variety of nutrients, including carbohydrates, proteins, fats, water, vitamins, and minerals. During the growth period, nutrition becomes essential for growth and development. Nutrients required for growth and development, such as protein, carbohydrates, fats, minerals, vitamins, and water, are required in nutrition [3]. If a person’s nutritional needs are not met, growth and development can be hampered. Today’s nutritional problems are the result of erroneous eating habits in which many people fail to pay attention to the diversity of food consumption, the body’s need for energy, and a balanced proportion of food. Food is one of life’s most basic needs [4]. This is because the body obtains the energy it requires for activities and metabolism by eating food. Eating food helps to keep the body healthy and the metabolism running smoothly [5].

Each type of food consumed has a different calorie content. Not only that, but each individual consumes a different amount of food. Many people nowadays eat far too much food [6]. Negative emotions, exposure to delicious food, an inability to restrain food intake, not feeling full, a craving for food, and even direct food addiction are all reasons for this excessive food consumption [7]. People suffering from diseases such as coronary heart disease, gout, and others must make careful food choices. Obesity, in addition to some of these diseases, is a problem for some people. Obesity is a disease that can be avoided in a variety of ways [8]. One of them is being in charge of recording and selecting. Another is keeping track of and selecting daily calorie consumption, as well as conducting a dietary assessment. If these foods are consumed uncontrollably, there will be an accumulation of excess calories in the body, leading to obesity [9]. Self-control in food consumption is required to avoid this, which includes measuring the calories of the food to be consumed. Controlling your diet can lower your risk of obesity and diseases, particularly diabetes (Table 1) [10].

Type 1 diabetes or insulin-dependent diabetes mellitus, type 2 diabetes or non-insulin-dependent diabetes mellitus, other types of diabetes mellitus, and gestational diabetes mellitus are the most common types of diabetes mellitus. Diabetes type 2 is a metabolic disorder characterized by hyperglycemia caused by insulin resistance and/or deficiency [11]. T2DM (Table 2) patients require diabetes management to properly and consistently control their blood glucose levels. If type 2 DM patients do not control their blood sugar levels properly, blood sugar levels can rise and fall, making them unstable, which can lead to complications. Diabetes mellitus control is accomplished through the application of basic diabetes mellitus control management principles, such as the modification of unhealthy lifestyles to become healthy through diet, physical exercise, and adherence to antidiabetic drug consumption [12].

As a result, an algorithm is required to make it easier for the system to recognize, compare, and study foods automatically using image data, so that differences between the foods to be consumed can be discovered [13]. At this time, it is undeniable that information technology is rapidly evolving and tends to aid humans in completing tasks more quickly and efficiently [14]. It is very possible to create a computation that can process information from an image or image for automatic object recognition in this all-digital era [15]. An “image processing” system is one in which an image is used as both the input and output to carry out the process. The goal of image processing is to improve an image’s quality so that it can be easily interpreted by humans or machines [16]. Because of its significant capabilities in modeling complex data such as images and sounds, deep learning has become one of the hottest topics in machine learning [17]. The convolutional neural network (CNN) is the deep learning method with the most significant results in image recognition right now [18].

CNN, or convolutional neural network, is a deep learning method that has been widely applied to image data [12]. In the case of image classification, the CNN method has succeeded in surpassing machine learning methods such as the SVM method, and currently has the most significant results in image recognition because CNN has a way of working that resembles the function of the human brain, where the computer will be given image data to study [19], trained to recognize each visual element on the image, and understand each image pattern, so that the computer will be able to identify the image [20]. CNN has recently advanced to become a sophisticated technique for image classification tasks [21]. CNN can investigate hierarchical structures. CNN can study the hierarchical features used for image classification, whereas the hierarchical approach can study more complex features with higher layers, resulting in a higher accuracy of the CNN method for image classification. CNN is said to be the most effective model for solving object detection and recognition problems [22]. On certain datasets, CNN research was able to perform digital image recognition with an accuracy that rivaled that of humans in 2012 [23].

In addition to CNN, there is the “You Only Look Once” (YOLO) algorithm. YOLO takes a different approach than the previous algorithm, which used a single neural network to process the entire image [24]. This network will divide the image into regions and then predict bounding boxes and probabilities; for each bounding region box, the probability of classifying as objects or not is weighed [25]. Detection is more complex than classification; classification can recognize objects but cannot tell where they are in the image [26]. Furthermore, if the image contains more than one object, the classification will fail. YOLO is a real-time detection smart neural network. YOLO has a simple architecture, namely a convolutional neural network [27]. This neural network uses only standard layer types: convolution with 3 × 3 kernels and max-pooling with 2 × 2 kernels [28]. The final convolutional layer employs 11 kernels to reduce the data to a 13x13x125 format. This 13 × 13 should look familiar; it is the size of the grid divided into images. In this case, 125 is the channel for each grid [29]. This 125 contains data for bounding boxes and class predictions. The reason for 125 is because each grid cell predicts 5 surrounding squares and is described by 25 data elements [30].

In this paper, we propose “Health to Eat: A Smart Plate with Real-Time Food Recognition, Classification, and Weight Measurement for Diabetic Patients’ Nutrition Control”, where this system will be useful to support or assist people with diabetes and non-diabetics in finding out information about the food they consume according to the nutritional standards they want to consume. Our research has a limitation: it is preliminary research (first stage), and we will use Korean food as our case study for our smart plate in this research. This research is related to our previous research, in which we developed a mobile-based diabetes application [31,32] that assists diabetic and non-diabetic sufferers and users in controlling their health activities through the use of a glucometer and exercise devices such as a treadmill and a connected gym cycle, which are integrated into a mobile-based diabetes application. In the future, we hope to be able to combine smart plates and mobile-based diabetes applications, with the hope that this system can further assist patients and users, where data from food consumption analysis will be processed to be used as information for users, so that doctors can use them as recommendations in order to manage diabetes.

## 2. Materials and Methods

### 2.1. Research Approach

Based on the research approach flow design (Figure 1), it is known that there are five stages carried out in this study.

IdentificationThe identification stage is the first stage in this study. The first step is to choose a research topic. The research in this case was conducted because of a lack of knowledge about food classification based on food names. Furthermore, the identification and formulation of the problem are carried out, where the researcher must know what the main issues that can be solved are and what can underpin the existence of this research after knowing the topic to be raised. The objectives are then determined and, in order to determine their own goals, there is a requirement for conformity with the results of the problem formulation. The objectives should be able to address any existing issues related to the topic raised. Following that, problem boundaries are established and a research methodology is developed to ensure that this research is directed toward the existing goals. Designing a research methodology will help researchers begin the steps and prepare to begin with the small things required by this research, such as the software used, the data used, how the data should be processed, and how the effective stages of analysis can then be explained coherently and clearly.Information GatheringThe stages of information gathering in this case are related to the process of studying literature; this stage is one that every researcher must complete, given the need to enrich references in the field that we will investigate. Looking at other research that has been done and is almost identical can help researchers develop and find other new things to explore.Data Validation and EvaluationThe third stage involves data validation and evaluation. Preprocessing is the process of improving data after they have been collected according to a predetermined topic and then corrected or prepared so that they can be processed by the algorithm.Hardware and Software DevelopmentAt this stage, smart plate hardware will be designed and developed in accordance with the appropriate components that have been analyzed for integration into a smart plate. The image processing process will be developed and designed on the software side, and the image captured by the camera will be processed on the software side, where the data will be processed and visualized in the form of food analysis results on a smart plate.TestingThe testing phase will be carried out to ensure the functionality of the designed and developed hardware and software. At this point, researchers will put the system to the test by putting various types of food on a smart plate.

### 2.2. Artificial Intelligence

Artificial intelligence (AI) is a technique used to solve problems by imitating the intelligence of living and non-living things. The study of how to make computers do things that humans currently do better is known as artificial intelligence. Artificial intelligence (AI) seeks to discover or model human thought processes in order to create machines that can mimic human behavior [33].

Artificial intelligence is the study, application, and teaching of computer programming to perform tasks that humans consider intelligent. As a result, it is hoped that computers will be able to mimic some of the functions of the human brain, such as thinking and learning. This artificial intelligence system can be trained or learned on the computer, a process known as “machine learning” [34].

Deep learning is a type of artificial neural network algorithm that takes metadata as input and processes them through a series of hidden layers of nonlinear transformation of the input data to calculate the output value. Deep learning algorithms have a distinct feature, namely the ability to extract automatically; this means that its algorithm can automatically capture the relevant features as needed in solving a problem. This type of algorithm is very important in artificial intelligence because it can reduce the programming burden by selecting explicit features. This algorithm can be used to solve problems in image recognition, speech recognition, text classification, and other applications that require supervision (supervised), no supervised (unsupervised), or some supervised (semi-supervised) [35].

Artificial neural networks, which are used in deep learning, mimic the operation of real neural networks. This algorithm employs hidden layer neurons to translate input data from the input layer to the target at the output layer. As the number of hidden layers’ increases, the algorithm becomes more complex and abstract. Deep learning neural networks are constructed by ascending through a simple hierarchy of several layers to a high level or many layers (multi-layer). Deep learning can be used to solve complex problems that have a large number of non-linear transformation layers based on this. Machine learning is an artificial intelligence approach that is widely used to replace or imitate human behavior in order to solve problems or perform automation. Machine learning, as the name suggests, attempts to mimic how humans and other intelligent creatures learn and generalize. Machine learning has at least two major applications: classification and prediction. CNNs, or convolutional neural networks, are the most popular neural network technique. CNN can process multidimensional data such as video and images. The operation of CNN is nearly identical to that of neural networks in general; the only significant difference is that it convolutes each unit in the CNN layer using a two-dimensional or high-dimensional kernel [36].

In CNN, the kernel is used to combine spatial features with a spatial form that is similar to the input medium. Then, CNN employs a variety of parameters to reduce the number of variables, making it easier to learn. The term “convolutional neural network” refers to the network’s use of a mathematical operation known as convolution. The CNN is then trained to examine the object’s features in order to predict it [37].

Feature learning (feature extraction layer); there is a layer in this section that is useful for receiving image input directly at the start and processing it to produce multidimensional array data output. This process has two layers: a convolution layer and a pooling layer, and each layer process produces feature maps in the form of numbers that represent images, which are then forwarded to the classification layer section.Classification layer; this layer is made up of several layers, each of which contains neurons that are fully connected to other layers. This layer receives input from the output layer of the feature learning section, which is then flattened with the addition of several fully connected hidden layers to produce output in the form of classification accuracy for each class.

### 2.3. YOLO Algorithm

The YOLO algorithm is a real-object detection algorithm that is currently being developed and has recently gained popularity. Most previous detection systems performed detection by applying a model to the image at multiple locations and scales and assigning a value to the image as material for detection. The You Only Look Once (YOLO) algorithm detects objects in real time [38]. A repurposed classifier or localizer is used as the detection system. A model is applied to an image at various scales and locations. The region with the highest image score will be considered a detection. An annotation process is required before beginning the training process to form the dataset. Each dataset has a class name, the object’s X and Y coordinates, and the length and width of the bounding box [39].

To detect objects in an image, YOLO employs an artificial neural network (ANN) approach. This network segments the image and predicts the bounding box and probability for each region. The predicted probabilities are then compared to the bounding boxes. YOLO has several advantages over a classifier-oriented system; it can be seen from the entire image at the time of the test, with predictions that are informed globally on the image. YOLO employs a convolutional-neural-network-like architecture. Only a convolution layer and a pooling layer are used by YOLO. It is adjusted for the final convolution layer based on the number of classes and prediction boxes desired. Convolutional neural networks, also known as CNNs or ConvNets, are a type of deep feed-forward artificial neural network widely used in image analysis. CNN is capable of detecting and recognizing objects in images. In general, CNN is not much different from a traditional neural network. CNN is made up of neurons with weight, bias, and activation functions. CNN is made up of an input layer, an output layer, and several hidden layers. The YOLO architecture is made up of only two layers: a convolution layer and a pooling layer [40].

The fifth-generation object detection model, YOLOv5, will be released in April 2020. In general, this model’s architecture is not significantly different from the previous YOLO generation. YOLOv5 is written in Python rather than C, as was the case in previous versions. It makes IoT device installation and integration easier [41].

Furthermore, the PyTorch community is larger than the darknet community, implying that PyTorch will receive more contributions and have greater growth potential in the future. Performance comparisons between YOLOv4 and YOLOv5 are difficult to make because they are written in two different programming languages on two different frameworks. However, after some time, YOLOv5 proved to be more performant than YOLOv4 in some cases, earning the computer vision community’s trust in addition to YOLOv4. There are several types of YOLOv5, each with its own detection speed and mAP performance [42] (Table 3).

### 2.4. Smart Plate Procedure

YOLO (You Only Look Once) is an object detection network. YOLOv5s is used in this study, and it is tasked with detecting objects, determining where on the image certain objects are present, and classifying these objects. Simply put, an image is used as input and the output is a bounding box vector and class prediction (Figure 2).

The food images used in this study were taken from the dataset. The number of images in the dataset is 30,800 images in jpg format, consisting of fifty different types of food. Furthermore, pre-processing of the data, which consists of labeling and changing the image size, is carried out. Image labeling is the initial stage where each image in the dataset is labeled with the aim of storing image information. The label process is carried out by giving a bounding box along with the class name for each image object. Next, the image size changes to improve the performance of the YOLOv5s model in object recognition.

The food images used in this study were taken from the dataset. The number of images in the dataset is 30,800 images in jpg format, consisting of fifty different types of food (Table 4). Furthermore, pre-processing of the data, which consists of labeling and changing the image size, is carried out. Image labeling is the initial stage where each image in the dataset is labeled with the aim of storing image information. The label process is carried out by giving a bounding box along with the class name for each image object. Next, the image size changes to improve the performance of the YOLOv5s model in object recognition.

The neural network used in this study for the YOLOv5s model consists of a convolutional layer with a 3 × 3 kernel and a maxpooling layer with a 2 × 2 kernel. The final convolutional layer employs a 1 × 1 kernel to shrink the data to a 13 × 13 40 form. The grid size is 13 x 13 and the sum of the filter formulas yields 40.

Real-time testing is carried out with an IMX219-160 camera module (waveshare). Tests are run to determine the level of object accuracy with a previously trained new model.

## 3. Results

### 3.1. System Development

Based on these concerns, the researchers proposed a study of measuring food calories with a Chenbo load cell weight sensor (1 kg) and a HX711 weight weighing A/D module pressure sensor. The IMX219-160 camera module (waveshare) and YOLOv5s are used in the study to detect the type of food as well as the load measuring sensor, specifically the loadcell sensor, to calculate the amount of food weight. The YOLOv5s algorithm is used to identify the type of food. This is because YOLOv5s is a classification method that uses training data to produce accurate results when a large amount of training data are used. The data training used up to 30,800 data points from 50 different types of foods across seven food categories (Table 3).

The results of food identification and weight measurement will be used to calculate the number of calories in food. In this study, a method for classifying food types is combined with the YOLOv5s algorithm, which is capable of classifying image data. Python 3.10.8 software with the Keras package aids in the processing of the YOLOv5s algorithm. The food images will be classified using the YOLOv5s algorithm, which will perform convolution operations on the data to form a pattern, and hardware and software will be integrated (Figure 3). MariaDB functions are used to process SQL data concurrently. MariaDB connects clients via TCP/IP, named pipes, or NT, as well as the built-in UNIX socket. Flask, on the other hand, serves as an application framework as well as a web display. Using Flask and the Python programming language, developers can easily create a structured web and manage web behavior. The results of the system’s analysis will be displayed on a web page.

### 3.2. Hardware and Software Design and Development

Figure 4, Figure 5, Figure 6 and Figure 7 depict the hardware design, which consists of a tool design and a system circuit design. Several electronic components will be used to build the system in the design of this hardware. The Raspberry Pi 4 model B, the Chenbo load cell weight sensor (1 kg), the HX711 weight weighing A/D module pressure sensor, and the IMX219-160 camera module are among the components used (waveshare). The camera module captures images of the tested samples, the loadcell sensor collects weight data, and the HX711 module is an ADC that converts the analog signal from the loadcell into a digital signal that the Raspberry Pi can process. The Raspberry Pi 4 model B is a microcontroller that processes the data obtained from the system input, which are then displayed as the system process’s final result.

Figure 7 depicts the implementation of the system that was created. This tool is shaped like a box, with a square board on top and an inverted angled board on the side. To take pictures, the camera module is mounted on the end of an inverted angled board, with the camera facing upwards. The loadcell sensor is installed between the box’s sides and the square board, which is used to place the food sample to be tested. Inside is a Raspberry Pi 4 model B and an IMX219-160 camera module (waveshare). All of these components will be linked to the Raspberry Pi 4 model B directly via the GPIO pins and the CSI socket.

Four Chenbo load cell weight sensors (1kg), HX711 weight weighing A/d module pressure sensors, and an IMX219-160 camera module (waveshare) are connected via jumper cables. The IMX219-160 camera module (waveshare) is linked via CSI to a Raspberry Pi 4 model B (Camera Serial Interface). The loadcell sensor is linked to the HX711 module, and the Raspberry Pi 4 model B is linked to the Chenbo load cell weight sensor (1 kg), HX711 weight weighing A/d module pressure sensor, and HX711 weight weighing A/d module pressure sensor.

The overall hardware (Raspberry Pi 4 I IMX219-160 camera module (waveshare), Chenbo load cell weight sensor (1 kg), and HX711 weight weighing A/d module pressure sensor) is connected via socket to the software (Server PC, YOLOv5s, MariDB, Flask); beginning with heavy data retrieval, image capture, and image data extraction, the YOLOv5s method processes until the system output results are displayed through the website display.

### 3.3. Experimental Results

The system flow must be implemented. To begin, the system’s library is initialized. The loadcell sensor should then be calibrated. If you press “enter” on the keyboard, the system will start. The loadcell sensor will then determine the weight of the food sample. The system will use the camera to capture an image, which will then be processed to extract color data from the image. Furthermore, food identification is based on the results of food image data extraction and existing training data using the YOLOv5s method. The system then calculates food calories based on the results of food identification and weight measurement (Table 5 and Table 6).

A formula for calculating food calories and calorie data samples for each food type is used. This test is performed to determine whether the system can identify food and calculate its calorific value. Figure 7 and Figure 8 depict the system results as they appear on the website page. Figure 9 depicts the system’s output, with the first display showing the type of food and its weight. The system identification process yields these types of food results, and the weight of the food yields loadcell sensor readings. Figure 8 depicts how the two systems appear on the website page. The second display in the second line will show the number of food calories calculated from the identification and reading of the loadcell sensor. Table 7, Table 8 and Table 9 shows the results of the accuracy of food identification using the YOLOv5s method, weight analysis and food nutrients in experimental foods.

### 3.4. Evaluation System

The total number of foods tested in this study was 50 and the total number of food images was 30,800. The tests were conducted on four different foods: rice, braised quail eggs in soy sauce, spicy beef soup, and dried radish. The researchers tested the accuracy of image detection, weight, and nutrition of the food three times with the same four food menus. The first test was carried out with a total of 800 images that focused on the four food menus tested; the second test increased the total number of foods to 50 with 10,000 images; and the third test added the same number of foods as in the second test but with an increased number of images, namely 20,000 (Figure 10).

Researchers obtained results for the detection of food types using three testing processes, with the highest accuracy value for each type of food being rice (58%), braised quail eggs in soy sauce (60%), spicy beef soup (62%), and dried radish (31%). In terms of weight and nutrition analysis, the system performed admirably, with 100% accuracy rates for rice, braised quail eggs in soy sauce, spicy beef soup, and dried radish.

## 4. Summary and Conclusions

Based on the findings of the analysis and testing, it is possible to conclude that the system can calculate the number of food calories measured using the YOLOv5s method and loadcell sensor readings. The identification process was carried out using the YOLOv5s method, which was based on training data that included up to 30,800 observations with 50 different types of food (Table 3). The use of variable k = 3 has the highest accuracy value of 62% in the YOLOv5s N method, namely the spicy beef soup food. The accuracy of the loadcell sensor readings performed demonstrates a zero percent error, indicating that the loadcell is capable of working with high accuracy while also providing good nutritional value.

Researchers can conclude from the results of the first, second, and third tests that there are variations in the value of accuracy, particularly in the detection of food types. We interpret this as being influenced by the increasing number of image types entered into the system, as well as the appearance of the food, because Korean food has a lot in common. This has an impact on the system’s ability to provide high accuracy values. As a result, the accuracy of the values in the first, second, and third tests varies. Our current research on how to design a smart plate has limitations.

This study is the first in a series on disease management for people with type 2 diabetes. We focused on the idea that the concepts we proposed could be applied on a small scale (amount and type of food), even with a simple model, in this study. The research we are conducting will be expanded upon in conjunction with our other research, specifically the development of a diabetes mobile application, with the ultimate goal of integrating all of these systems into a single diabetes healthcare management system that will benefit both diabetic and non-diabetic patients.

## 5. Future Work

Our project’s mobile-based diabetes application will be integrated with smart plates in the future. In the future, patients and users with diabetes or without diabetes will be able to easily control their health, from diet and activity to medication and even using a glucometer, thanks to smart plates and mobile-based diabetes applications.

## Figures and Tables

**Figure 1 sensors-23-01656-f001:**
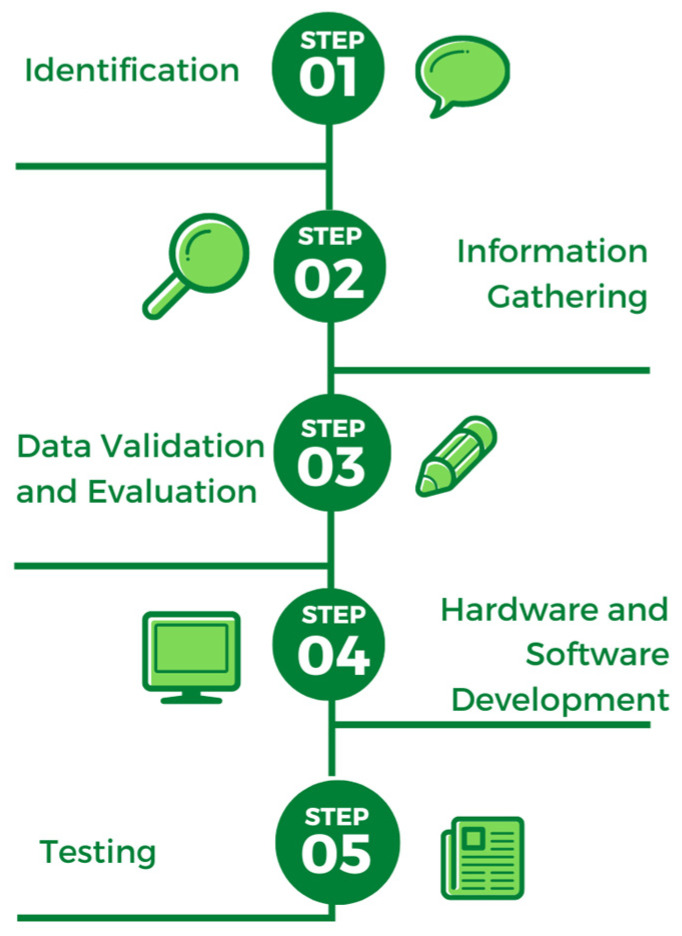
Research approach.

**Figure 2 sensors-23-01656-f002:**
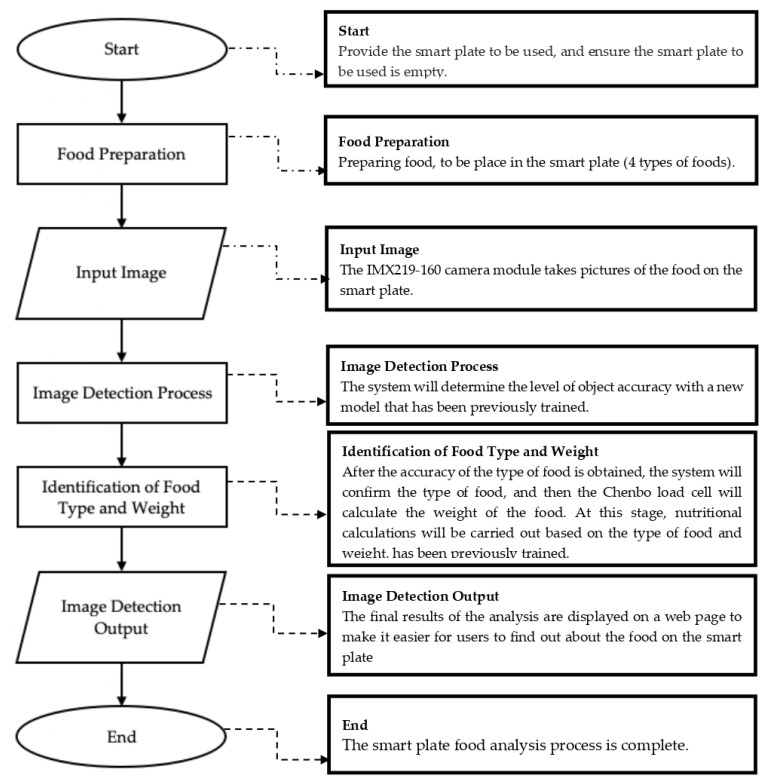
Flowchart of a smart plate.

**Figure 3 sensors-23-01656-f003:**
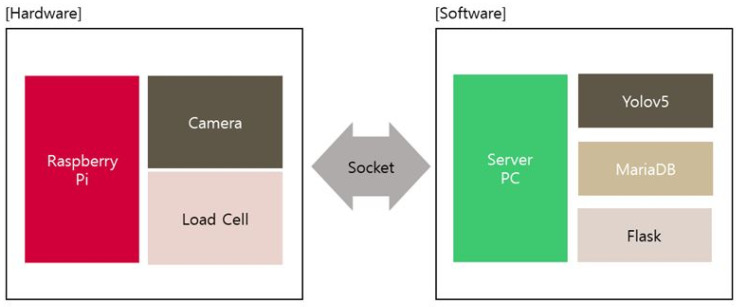
Hardware and software integration.

**Figure 4 sensors-23-01656-f004:**
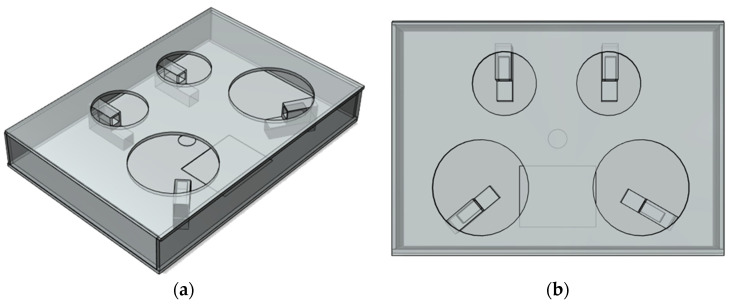
(**a**) Design of smart plate side view design, (**b**) design of smart plate top view design.

**Figure 5 sensors-23-01656-f005:**
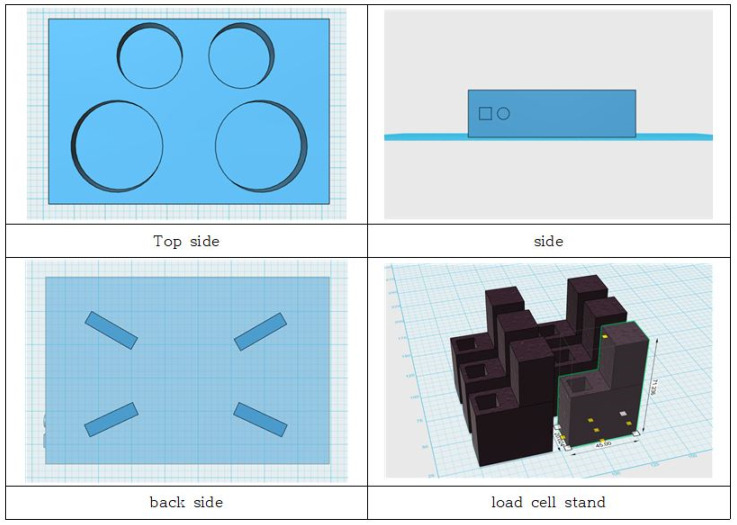
3D design of smart plate components.

**Figure 6 sensors-23-01656-f006:**
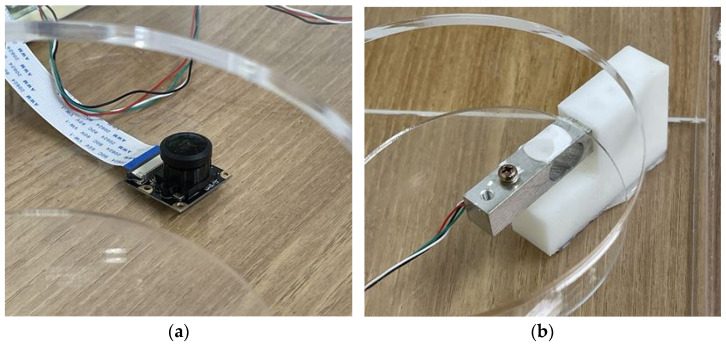
(**a**) Camera, (**b**) load cell.

**Figure 7 sensors-23-01656-f007:**
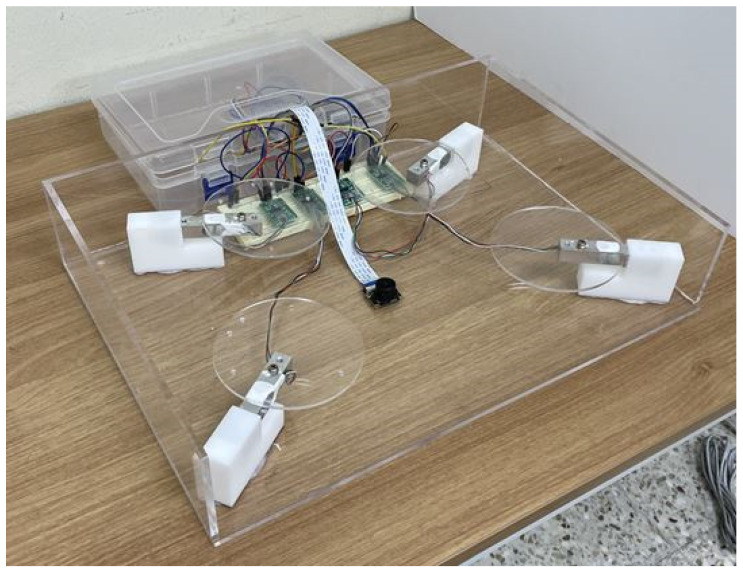
Smart plate.

**Figure 8 sensors-23-01656-f008:**
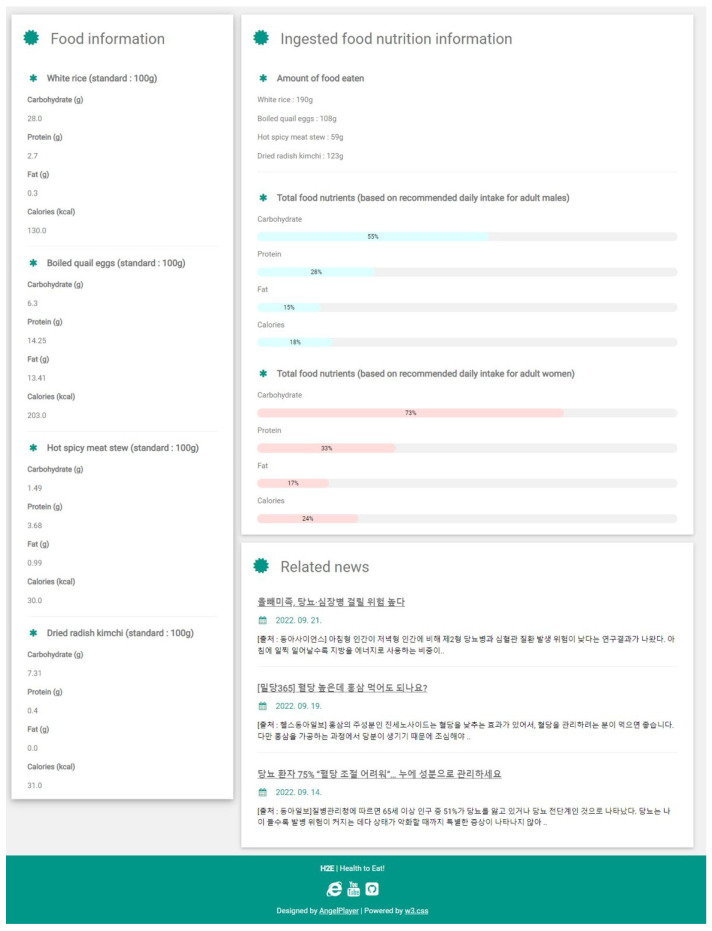
Data visualization.

**Figure 9 sensors-23-01656-f009:**
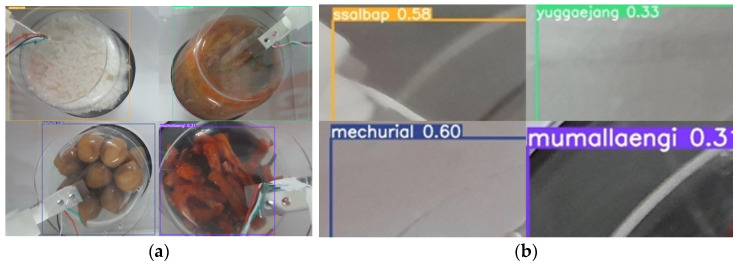
(**a**) Capturing image, (**b**) zoomed-in image.

**Figure 10 sensors-23-01656-f010:**
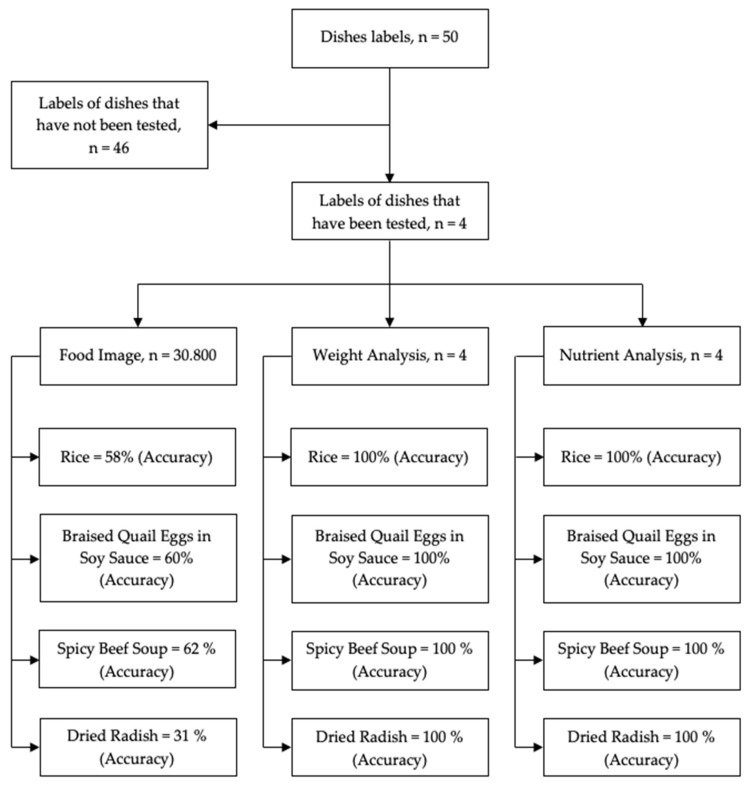
Evaluation results of the system.

**Table 1 sensors-23-01656-t001:** Risk factor.

No	Risk Factor	
	Factor	Description
1	Obesity (overweight)	There is a significant link between obesity and blood sugar levels; the degree of obesity with body mass index (BMI) > 23 can lead to an increase in blood glucose levels of up to 200 mg%.
2	Hypertension	An increase in blood pressure beyond the normal range of hypertensive patients is closely associated with improper storage of salt and water or increased pressure in the body of the peripheral vascular system.
3	Dyslipidemia	Dyslipidemia is a condition characterized by elevated blood fat levels (triglycerides > 250 mg/dL). There is a relationship between an increase in plasma insulin and low high-density lipoprotein (HDL) (<35 mg/dL).
4	Age	Individuals aged >40 years are susceptible to DM, although it is possible for individuals aged <40 years to avoid DM. The increase in blood glucose occurs at the age of about 45 years and the frequency increases with age.
5	Genetic	Type 2 DM is thought to be associated with familial aggregation. The empirical risk in the event of type 2 DM will increase two to six times if there are parents or family members suffering from type 2 DM.
6	Alcohol and Cigarettes	An individual’s lifestyle is associated with an increase in the frequency of type 2 DM. Most of this increase is associated with increased obesity and decreased physical activity, while other factors associated with the shift from a traditional to a westernized environment, including changes in cigarette and alcohol consumption, also played a role in the increase. Type 2 DM alcohol will interfere with blood sugar metabolism, especially in people with type 2 DM, so it will complicate regulation and increase blood sugar.

**Table 2 sensors-23-01656-t002:** List of abbreviations.

No	Abbreviation	Meaning
1	T2DM	Type 2 diabetes mellitus
2	DM	Diabetes mellitus
3	AI	Artificial intelligence
4	CNN	Convolutional neural network
5	YOLO	You Only Look Once
6	ANN	Artificial neural network

**Table 3 sensors-23-01656-t003:** Image identification accuracy test.

No	YOLOv5								
	Model	Size (Pixels)	mAPval 0.5:0.95	mAPval 0.5	Speed CPU b1 (ms)	Speed V100 b1 (ms)	Speed V100 b32 (ms)	Params (M)	FLOPs @640 (B)
1	YOLOv5n	640	28.4	46.0	45	6.3	0.6	1.9	4.5
2	YOLOv5s	640	37.2	56.0	98	6.4	0.9	7.2	16.5
3	YOLOv5m	640	45.2	63.9	224	8.2	1.7	21.2	49.0
4	YOLOv5l	640	48.8	67.2	430	10.1	2.7	46.5	109.1
5	YOLOv5x	640	50.7	68.9	766	12.1	4.8	86.7	205.7

**Table 4 sensors-23-01656-t004:** List of foods.

No	Food	
	Category	Name
1	Rice	Rice
2	Soup	Porridge
3	Soup	Vegetable Porridge
4	Meat	Chicken Skewers
5	Meat	Pork Galbi
6	Meat	Smoked Duck
7	Meat	Chicken Wings
8	Vegetable	Grilled Deodeok
9	Pancake	Seafood and Green Onion Pancake
10	Meat	Pan-Fried Battered Meatballs
11	Pancake	Pan-Fried Battered Summer Squash
12	Meat	Omelet Roll
13	Fish	Stir-Fried Anchovies
14	Fish	Stir-Fried Fishcake
15	Meat	Stir-Fried Sausages
16	Rice	Tteokbokki
17	Fried	Braised Quail Eggs in Soy Sauce
18	Fish	Deep-Fried Loach
19	Fish	Deep-Fried Filefish Jerky
20	Meat	Deep-Fried Chicken
21	Meat	Pork Cutlet
22	Meat	Deep-Fried Chicken Gizzards
23	Fried	Deep-Fried Potatoes
24	Vegetable	Deep-Fried Laver Roll Stuffed with Glass Noodles
25	Vegetable	Deep-Fried Vegetables
26	Vegetable	Pickled Radish Salad
27	Vegetable	Dried Radish
28	Vegetable	Bean Sprout
29	Vegetable	Julienne Radish Fresh Salad
30	Fish	Sea Snail Salad
31	Vegetable	Laver Salad
32	Vegetable	Japchae
33	Vegetable	Diced Radish Kimchi
34	Fish	Soy Sauce Marinated Crab
35	Vegetable	Garlic Stem Salad
36	Vegetable	Pickled Perilla Leaf
37	Vegetable	Pickled Radish
38	Fish	Sliced Raw Salmon
39	Rice	Rice Cake Stick
40	Rice	Rice Cake with Honey Filling
41	Rice	Steamed Rice Cake
42	Rice	Buckwheat Crepe
43	Rice	Rainbow Rice Cake
44	Rice	Snow White Rice Cake
45	Rice	Half-Moon Rice Cake
46	Rice	Bean-Powder-Coated Rice Cake
47	Rice	Honey Cookie
48	Rice	Fried Rice Sweet
49	Rice	Sweet Rice Puffs
50	Soup	Spicy Beef Soup

**Table 5 sensors-23-01656-t005:** Total of food images.

No	Food Image	
	Experiment	Total
1	First Experiment Test 4 types of food with 200 images each	800 images
2	Second Experiment Test 50 types of food with 200 images each	10,000 images
3	Third Experiment Test 50 types of food with 400 images each	20,000 images
	Total Images Three Experiments	30,800 images

**Table 6 sensors-23-01656-t006:** List of experimental foods.

No	Food	
	Category	Name
1	Rice	Rice (쌀밥)
2	Stewed foods	Braised Quail Eggs in Soy Sauce (메추리알)
3	Soups	Spicy Beef Soup (육개장)
4	Vegetable	Dried Radish (무말랭이)

**Table 7 sensors-23-01656-t007:** Image identification accuracy test.

No	Model				
	Experiment	Food 1 (%)	Food 2 (%)	Food 3 (%)	Food 4 (%)
1	Test 4 types of food with 200 images each	0.49	0.39	0.62	0.20
2	Test 50 types of food with 200 images each	0.50	0.55	0.34	0.25
3	Test 50 types of food with 400 images each	0.58	0.60	0.33	0.31

**Table 8 sensors-23-01656-t008:** Weight analysis.

No	Smart Plate				
	Experiment	Plate A	Plate B	Plate C	Plate D
1	1	Rice	Braised Quail Eggs in Soy Sauce	Spicy Beef Soup	Dried Radish
	Amount of food served (g)	190	108	58	123
2	2	Rice	Braised Quail Eggs in Soy Sauce	Spicy Beef Soup	Dried Radish
	Amount of food served (g	190	108	58	123
3	3	Rice	Braised Quail Eggs in Soy Sauce	Spicy Beef Soup	Dried Radish
	Amount of food served (g)	190	108	58	123
**Relative Error Ratio (%)** **(Error Ratio = (Actual Weight − Measured Weight): Actual Weight)**		0%	0%	0%	0%

**Table 9 sensors-23-01656-t009:** Food nutrients in experimental foods.

No	Food Nutrient (per 100 g)				
	Name	Carbohydrate (g)	Protein (g)	Fat (g)	Kcal
1	Rice	28.0	2.7	0.3	130.0
2	Braised Quail Eggs in Soy Sauce	6.3	14.25	13.41	203.0
3	Spicy Beef Soup	1.49	3.68	0.99	30.0
4	Dried Radish	7.31	0.4	0.0	31.0

## Data Availability

All data are contained within the article or https://www.mdpi.com/2076-3417/11/5/2006, https://www.mdpi.com/2079-9292/10/15/1820.

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
