# Peer review of "Health to Eat: A Smart Plate with Food Recognition, Classification, and Weight Measurement for Type-2 Diabetic Mellitus Patients’ Nutrition Control"

_sensors, 2023, doi:10.3390/s23031656_

Round 1

Reviewer 1 Report

General comment:

The proposed system (smart plate) is cogent and compelling, particularly in the field of food intake research. However, the current version of the proposed system is very little into its implementation. Additionally, my major critic on the manuscript is that the writing style is not scientific, more like a thesis report rather than a journal article. Many of the sentences are vague (and long), difficult to follow. I have highlighted (in yellow) few of the sentences where the author should rewrite/revise accordingly.

Specific comment:

1)    Abstract: “This study involved 50 types of food with a total of 38,000 foods”- is it correct? 38,000 foods or 30,800 foods? The authors should be consistent throughout the paper about this number.

2)    Abstract: the last sentence says that “nutrition (100%)” – is this accuracy for nutrition of all 50 types of foods?

3)    Why reference [19] is coming right after reference [2]? The reference should be organized and cited in sequence.

4)    Not enough literature review is performed. Related works on smart plate for food recognition must be mentioned along with current research trends, challenges, and limitations.

5)    The research method in figure 1 should be explained in the light of the proposed work. Currently, the research method is described very generally.

6)    The section 2.2 should be revised. The section reads like accumulation of information without any logical transitions between paragraphs.

7)    Line 267: which table 5 is referred here?

8)    Please provide more information about the columns of Table 2.

9)    Line 293: The authors are claiming to test the system in real-time environment. Please produce some results.

10)          Please provide some discussions about MariaD8 and Flask in Figure 3.

11)          Lines 365-366: “The equation for calculating food calories and calorie data samples for each type of food.” – Please write the equations.

12)          How your model would identify the sauces and/or liquid intake?

13)          Please write the formula of relative error ratio for Table 7.

14)          Please provide sufficient discussion on the confusion matrix given in Figure 10. And please rewrite in food type English.

15)          Figure 11 seems to be redundant. Please justify with proper explanation.

16)          No comparison is made between the proposed method's results and the existing state-of-the-art methods' results. Comparison should be presented at least with two works.

17)          The obtained results are not discussed or explained properly. 

18)          Please mention the limitations of the work. 

Author Response

Response to Reviewer 1 Comments

Point 1: Abstract: “This study involved 50 types of food with a total of 38,000 foods”- is it correct? 38,000 foods or 30,800 foods? The authors should be consistent throughout the paper about this number.

Response 1: The correct is 30.800 Total (Food Images), Reference Table 4. Total of Food Image.

Note: Updated

Point 2: Abstract: the last sentence says that “nutrition (100%)” – is this accuracy for nutrition of all 50 types of foods?

Response 2: For Experiment to see the Nutrition, we only evaluate 4 Types of Food, Reference Table 8. Food Nutrient Experimental Food.

Point 3: Why reference [19] is coming right after reference [2]? The reference should be organized and cited in sequence.

Response 3: Reorganized the reference start from 1

Point 4: Not enough literature review is performed. Related works on smart plate for food recognition must be mentioned along with current research trends, challenges, and limitations.

Response 4:  We also accept this condition. Because it’s very hard to find a good quality of research related with our work.

  • In our paper we bring; 9 Papers related to Image/ Food Recognition; 1 Paper related to smart plate; 10 Papers about Smart Detection;
  • We found some paper with who discussed about smart plate, but they just publish in a blog without any information about the publisher.
  • There are some Key Word of Smart Plate if we want to search, we can find some research but in the field of Civil Engineering, or structural engineering. They have a research topic also with key word of (Smart Plate).

Point 5: The research method in figure 1 should be explained in the light of the proposed work. Currently, the research method is described very generally.

Response 5: We updated the name to research approach, because this work is a part of one research project. Which is the other one is (Mobile Diabetes Application)

Point 6: The section 2.2 should be revised. The section reads like accumulation of information without any logical transitions between paragraphs.

Response 6: 

Updated section 2

2.1 Research Approach

2.2 Artificial Intelligence

2.3 YOLO Algorithm

2.4 Smart Plate Procedure

Point 7: Line 267: which table 5 is referred here?

Response 7: Updated, this referred to Table 2. Image Identification Accuracy Test.

Point 8: Please provide more information about the columns of Table 2.

Response 8: Table 2 is the explanation of Types of YOLOv5 (YOLOv5n, YOLOv5s, YOLOv5m, YOLOv5l, YOLOv5x), in terms of size, mAPval 0.5:0.95, mAPval 0.5, Speed CPU b1(ms), Speed V100 b1 (ms), Speed V100 b32 (ms), Params (M) and FLOPs @640 (B)

Point 9: Line 293: The authors are claiming to test the system in real-time environment. Please produce some results.

Response 9: 

Figure 9. (a) Capturing Image, (b) Zoom in Image.

Figure 8. Data Visualization.

Point 10: Please provide some discussions about MariaD8 and Flask in Figure 3.

Response 10: Updated

Point 11: Lines 365-366: “The equation for calculating food calories and calorie data samples for each type of food.” – Please write the equations.

Response 11:  We are not mention in the paper about the Nutritional database, but actually in our study we are using Database Version 7 which is contains of 2963 Nutritional Data (Divided to 6 Groups: Grain, Meat (Low, Medium and High Fat), Vegetables, Fat and Oil, Milk and Fruit) from Korean Diabetes Association. Based on this Nutritional Data, we applied to our work.

This is the step how this System calculate the Calories:

  1. Analyze food (Identify Type of Food)
  2. Measure the Weight of Food
  3. Calculate the Calories referred to Database V7 (Nutritional Database). The formula:

Measured Weight x Calorie (DB 7) = Number of Calorie

Point 12: How your model would identify the sauces and/or liquid intake?

Response 12: Our Study is a preliminary study; we have a lot of limitation. We just focused on how our first study can at least be applied in some category of Food. Our future work is to integrated this smart plate with our current work (Mobile Diabetes Application)

Point 13: Please write the formula of relative error ratio for Table 7.

Response 13:  Updated in Table 7

Error Ratio = (Actual Weight - Measured Weight): Actual Weight

Note: Actual Weight (in Database)

Point 14: Please provide sufficient discussion on the confusion matrix given in Figure 10. And please rewrite in food type English.

Response 14: We decided to delete the image, It’s hard to change because our labelling process is done by using Korean food name as our case study. We are really appreciating about your suggestion, but we have a short time to submit final report to our research fund. 

Point 15: Figure 11 seems to be redundant. Please justify with proper explanation.

Response 15: As you can see the flow, Figure 11 Explain 3 Different Category, Food Image, Weight Analysis and Nutrient. Item Weight and Nutrient looks redundant because this experiment is in the same process. If Weight will impact number of Nutrient. 

Point 16: No comparison is made between the proposed method's results and the existing state-of-the-art methods' results. Comparison should be presented at least with two works.

Response 16:  We also accept this condition. Because it’s very hard to find a good quality of research related with our work.

  • In our paper we bring; 9 Papers related to Image/ Food Recognition; 1 Paper related to smart plate; 10 Papers about Smart Detection;
  • We found some paper with who discussed about smart plate, but they just publish in a blog without any information about the publisher.

There are some Key Word of Smart Plate if we want to search, we can find some research but in the field of Civil Engineering, or structural engineering. They have a research topic also with key word of (Smart Plate).

And this work is actually based on research fund, and for social work in Samcheok City, Korea

Point 17: The obtained results are not discussed or explained properly.

Response 17:  This work is actually a preliminary work, so the main result is our design (Propose Model of Smart Plate) and some experimental result to make sure our design can be implemented:

3.1. System Development

3.2. Hardware and Software Design and Development

3.3. Experimental Results

3.4. Evaluation System

Point 18: Please mention the limitations of the work.

Response 18: Part 4. Summary and Conclusion

Note:

Our paper actually already accepted at Applied Sciences, but our one Author Prof Je-Hoon Lee as a Reviewer at Sensor, so they invited us to submit at sensor. Our paper also already checked (English) by Professor Dr. Tini Mogea from Manado State University (tinimogea@unima.ac.id) as a Professor from English Department. 

Previous Publication

  • https://www.mdpi.com/2076-3417/13/3/1251
  • https://www.mdpi.com/2076-3417/13/1/8 

Reviewer 2 Report

Prima facie, I found it difficult to understand the study with current at single glance, in my opinion change title as following ´Smart Plate for Type-2 Diabetes Mellitus Patients' Nutrition Control: Food Recognition, Classification, and Weight Measurement´.

I suggest removing Korean transcript from Table 3 showing List of Food since MDPI is English language publisher and may misunderstood for regional  language bias.

Abstract: The management of Type 2 diabetes mellitus (T2DM) is generally only focused on pharmacological therapy.

Please edit above sentence as this is not true ´Only´ pharmacological intervention are in practice.

Line 24, correct:  (waveshare…………

Line 86-87, cite a very recent paper with https://doi.org/10.1038/s41598-022-16828-6 to update reference 6 with the latest development.

Please cut down introduction as it is too long, I suggest completely remove nutritional aspect (opening paragraph), diabetes’s classification. Describe briefly problem the that T2DM nutritional maintenance is a big health issue and needs to considered for modern tool development to improve health in this context.

Section: Information Gathering, I strongly suggest authors to look into nutritional related databases for T2DM rather than  relying on published body of literature and include here in the section authors mentioned. Databases have been instrumental in developing computation and in silico method for cutting edge ways to fight chronic conditions.

Figure 2. Flowchart of Smart Plate., please provide details of each step either in schematic or in figure captions. Readers often cannot refer to the text for such details and figures/captions are right places to add details.

Line 277, ´The food images used in this study were taken from the dataset´, which dataset authors are talking about?

English grammar and style are of poor quality. Many punctuation errors, typos and banal phrasing makes it difficult to follow. Paper need revision with native English speakers. Some are reported below

Line 51, typo delicious

 Line 58, causing obesity and obesity, again line 61, line 71 repeat of words

Line 180 Artificial intelligence (artificial intelligence) rather then Artificial intelligence (AI)

Page 4, line 89-94, sentence is overly long, please split and cite a latest report DOI: 10.1109/ACCESS.2022.3233110 with sentence ending with ´……………..because CNN 91 has a way of working that resembles the function of the human brain ´ to support he statement.

Line 230 – 239, You only look once (YOLO) need to be in the beginning of paragraph.

Figures should be self-explanatory. All figures are with insufficient descriptions. Figure 4. (a) Side View Design, (b) Top View Design, add description of all components shown therein.

Provide a list of all abbreviations used in this study.

Author Response

Response to Reviewer 2 Comments
Point 1:  Food Recognition, Classification, and Weight Measurement´. I suggest removing Korean transcript from Table 3 showing List of Food since MDPI is English language publisher and may misunderstood for regional  language bias.
Response 1: Updated

Point 2: Abstract: The management of Type 2 diabetes mellitus (T2DM) is generally only focused on pharmacological therapy. Please edit above sentence as this is not true ´Only´ pharmacological intervention are in practice.

Response 2: Updated.

Point 3: Line 24, correct:  (waveshare…………

Response 3: Updated: the correct is (waveshare)

Point 4:  Line 86-87, cite a very recent paper with https://doi.org/10.1038/s41598-022-16828-6 to update reference 6 with the latest development.

Response 4: Already replace reference 6 with https://doi.org/10.1038/s41598-022-16828-6

Point 5: Please cut down introduction as it is too long, I suggest completely remove nutritional aspect (opening paragraph), diabetes’s classification. Describe briefly problem the that T2DM nutritional maintenance is a big health issue and needs to considered for modern tool development to improve health in this context.

Response 5: Is it possible to keep this part? Because the purpose of this research is to be a final output for our research Fund. In this case we have other work (Mobile Diabetes), already published. In the future we will integrate this mobile app and this smart plate project.

Point 6: Section: Information Gathering, I strongly suggest authors to look into nutritional related databases for T2DM rather than relying on published body of literature and include here in the section authors mentioned. Databases have been instrumental in developing computation and in silico method for cutting edge ways to fight chronic conditions.

Response 6: We are not mention in the paper about the Nutritional database, but actually in our study we are using Database Version 7 which is contains of 2963 Nutritional Data (Divided to 6 Groups: Grain, Meat (Low, Medium and High Fat), Vegetables, Fat and Oil, Milk and Fruit) from Korean Diabetes Association. Based on this Nutritional Data, we applied to our work. 

Point 7: Figure 2. Flowchart of Smart Plate., please provide details of each step either in schematic or in figure captions. Readers often cannot refer to the text for such details and figures/captions are right places to add details.

Response 7: Updated Figure with Explanation

Point 8: Line 277, ´The food images used in this study were taken from the dataset´, which dataset authors are talking about?

Response 8: Food dataset based on list of Food in Table 3. List of Food

Point 9: 

English grammar and style are of poor quality. Many punctuation errors, typos and banal phrasing makes it difficult to follow. Paper need revision with native English speakers. Some are reported below

Line 51, typo delicious

Line 58, causing obesity and obesity, again line 61, line 71 repeat of words

Line 180 Artificial intelligence (artificial intelligence) rather then Artificial intelligence (AI)

Page 4, line 89-94, sentence is overly long, please split and cite a latest report DOI: 10.1109/ACCESS.2022.3233110 with sentence ending with ´……………..because CNN 91 has a way of working that resembles the function of the human brain ´ to support he statement

Response 9: 

Line 51 (Updated)

Line 58 (Updated)

Line 61 (Updated)

Line 71 (Updated)

Line 180 (Updated)

Page 4 line 89-94 (Updated) using: 10.1109/ACCESS.2022.3233110 with sentence ending with

Point 10: 

Line 230 – 239, You only look once (YOLO) need to be in the beginning of paragraph.

Figures should be self-explanatory. All figures are with insufficient descriptions. Figure 4. (a) Side View Design, (b) Top View Design, add description of all components shown therein.

Provide a list of all abbreviations used in this study.

Response 10: 

Updated section 2

2.1 Research Approach

2.2 Artificial Intelligence

2.3 YOLO Algorithm

2.4 Smart Plate Procedure

We updated the name of figure 4, Figure 4. (a) Design of Smart Plate, Side View Design, (b) Design of Smart Plate, Top View Design.

The Description of the components at Figure 5

List of Abbreviation, Table 2. List of Abbreviation

Note:

Our paper actually already accepted at Applied Sciences, but our one Author Prof Je-Hoon Lee as a Reviewer at Sensor, so they invited us to submit at sensor. Our paper also already checked (English) by Professor Dr. Tini Mogea from Manado State University (tinimogea@unima.ac.id) as a Professor from English Department. 

Previous Publication :

  • https://www.mdpi.com/2076-3417/13/3/1251 
  • https://www.mdpi.com/2076-3417/13/1/8 

Round 2

Reviewer 1 Report

The authors have responded adequately to most of the questions and acknowledged the limitations of their study.  Since the authors acknowledge "Our Study is a preliminary study; we have a lot of limitation. " therefore, a separate section (or paragraph) for the limitations of the work should be included.

Author Response

Response to Reviewer 1 Comments

Point 1: 

Response 1: Update (Page 2 lines 116-130)

In this paper, we propose "Health to Eat: A Smart Plate with Real-Time Food Recognition, Classification, and Weight Measurement for Diabetic Patients' Nutrition Control," where this system will be useful to support or assist people with diabetes and non-diabetics in finding out information about the food they consume according to the nutritional standards they want to consume. Our research has a limitation: it is preliminary research (first stage), and we will use Korean food as our case study for our smart plate in this research. This research is related to our previous research, in which we previously developed a mobile-based diabetes application [31, 32] that assists diabetic and non-diabetic sufferers and users in controlling their health activities through the use of a glucometer and exercise devices such as a treadmill and a connected gym cycle, which are integrated into a mobile-based diabetes application. In the future, we hope to be able to combine smart plates and mobile-based diabetes applications, with the hope that this system can further assist patients and users, where data from food consumption analysis will be processed to be used as information for users, so that doctors can use them as recommendations in order to manage diabetes.

Reviewer 2 Report

accept

Author Response

Response to Reviewer 1 Comments

Point 1: I would like to sign my review report

Response 1:

Thank you Professor for all the suggestions and comments
